# Threats Posed to the Rediscovered and Rare *Salvia ceratophylloides* Ard. (Lamiaceae) by Borer and Seed Feeder Insect Species

**Carmelo Peter Bonsignore** [1,*], **Valentina Lucia Astrid Laface** [2], **Gregorio Vono** [2], **Rita Marullo** [2], **Carmelo Maria Musarella** [2] and **Giovanni Spampinato** [2]

1 Laboratorio di Entomologia ed Ecologia Applicata (LEEA), Dipartimento Patrimonio, Architettura, Urbanistica, Università Mediterranea di Reggio Calabria, Via dell'Università, 89124 Reggio Calabria, Italy
2 Dipartimento di Agraria, Università Mediterranea di Reggio Calabria, Località Feo de Vito, 89124 Reggio Calabria, Italy; vla.laface@unirc.it (V.L.A.L.); gregorio.vono@unirc.it (G.V.); rmarullo@unirc.it (R.M.); carmelo.musarella@unirc.it (C.M.M.); gspampinato@unirc.it (G.S.)
* Correspondence: cbonsignore@unirc.it; Tel.: +39-0965-1696318

**Abstract:** The effects of herbivorous insects on a plant population are not always well tolerated. This is especially true if the herbivorous actions are directed toward rare plant species. *Salvia ceratophylloides* Ard. is a rare endemism of southern Italy. Observations of the plants in situ revealed that many of them were under severe stress and did not produce seeds. Therefore, to find out which factors affect the reproductive activity as a whole, an observational study was carried out. We found bottom-up and top-down effects on plant health and reproduction associated with herbivorous action. *Squamapion elongatum* (Coleoptera, Curculionoidea, Apionidae), in all monitored sites, infested plants non-uniformly but was able to threaten the health condition, flowering, and seed production of sage by digging tunnels into the sage branches (bottom-up action), and then secondarily by seed feeder *Systole salvia* Zerova (Hymenoptera, Eurytomidae) predating sage seeds (top-down action). Mainly, chalcid parasitoid wasps such as *Trichomalus* spp. (Hymenoptera, Pteromalidae), as well as *Eupelmus vesicularis* and *E. muellneri* (Hymenoptera, Eupelmidae), limited the herbivorous *S. elongatum* population and the seed herbivore *S. salviae* emerged with its parasitoid *Ormyrus diffinis* (Hymenoptera, Ormyridae). Overall, this study showed how ecological interactions among herbivores, their host, and their natural enemies act on this sage species in all sites investigated. Among the herbivores, mainly *S. elongatum* affected this rare sage species, which should be taken into consideration, especially in the formulation of biological control solutions and for improving operating practice aimed at reproducing the species. This study provides the molecular characterization of the herbivorous species involved, in order to support future projects to evaluate the intra- and interspecific genetic variability of insects, their evolutionary relationships, and phylogeny studies.

**Keywords:** sage; *Salvia ceratophylloides*; Apionidae; *Systole*; ecological function; herbivore; rare species; diversity; seed

## 1. Introduction

The loss of both animal and plant biodiversity is basically related to several key factors, some of which are related to human activity (fragmentation and loss of habitat, pollution, etc.) and others related to climatic events and geological processes [1–3]. Often, the processes that determine the recovery of biodiversity are reversible and are linked to the reduction of pollutants and an increase in habitat favorable to the species [4]. To achieve this aim, it is often essential to know which factors at the local scale contribute to regulating the relationships among plants, the environment, and herbivores [5,6]. Among the biotic factors, the effects of both native and non-native herbivores must be carefully considered, particularly the herbivorous species capable of affecting the host plant's vitality and production of reproductive structures, even if these aspects are not always attentively evaluated [7,8]. In rare and endemic species, the reproductive aspects are very important,

as their populations are particularly restricted, as is their spread [9,10]. Reproductive strategies, therefore, are essential for determining the quantity and quality of offspring [11,12], which can be conditioned by the action of some herbivores or pathogens limiting the fitness of a species [13–15]. The detrimental action by herbivores, when occurring in rare species on a small number of individuals/populations, directly affects both suitability and genetic composition, influencing the population dynamics and long-term persistence [16]. *Salvia ceratophylloides* Ard. is strict endemic species that grows only in the lower belt of Aspromonte in the surroundings of Reggio Calabria [17,18]. Until 2008, *S. ceratophylloides* was considered a species "extinct in the wild" (EW) by authors who studied the genus *Salvia* in Italy, as well different floristic databases and checklists [19–22]. The intense urbanization of the areas surrounding the city of Reggio Calabria has led to profound changes in the landscape, with a decisive fragmentation and loss of natural habitats. This probably caused the extinction of the species from the sites in which researchers were attempting to collect it prior to 1900 [23]. However, Spampinato [24] did not exclude that some populations could still have survived on some sandy hills of Reggio Calabria. Further field surveys carried out in the last decade have allowed us to verify the presence of *S. ceratophylloides* in other areas and to discover some new populations in Reggio Calabria (Puzzi, Cataforio) [25–28] so as to better define its taxonomic and ecological characteristics, as well as its conservation status [18,28]. Since 2018, it has been observed that both sage plants and their seeds are infested by herbivores, thereby affecting their growth, inflorescence formation, and seed germination. Understanding which of the biotic causes can compromise the vitality and the reproductive capacity of this sage species and its performance in the area of diffusion has become essential. To accomplish this aim, we conducted non-manipulative experiments both in the field and laboratory. Specifically, in the field, we differentially evaluated the plant species' vitality according to the direct effects of the borer on the reproductive structures (inflorescence formation), while indirectly, in the laboratory, we investigated the sage's seed predator insect. Moreover, we evaluated the occurrence of natural enemies preventing herbivorous damage. The obtained results constitute the first contribution on the insect species related to the endemic *S. ceratophylloides*, generating new questions for future research on arthropod–plant interactions.

## 2. Materials and Methods

### 2.1. Study Species

*S. ceratophylloides* is an herbaceous plant (scapose hemicryptophyte) (Figure 1a), densely pubescent for both glandular and simple patent hairs. The plant is 30–90 cm high, has upright or ascending stems, normally lignified and much ramified at the base. The inflorescences consist of several verticillasters (5–6), each with 4–6 flowers that are typically bilabiate (Figure 1b), and corolla violet (three times the calyx) with an upper lip strongly bent to cap the pubescent glandular stamens on the outside. The fruit is a peculiar schizocarp: a microbasarium made up of four dark brown, spherical–ovoidal one-seeded mericarps (nutlets) with a thickened margin. *S. ceratophylloides* has a main flowering period in spring, from April to June, and a second flowering period in autumn, from October to November. The fruiting occurs a few weeks after flowering; the species is a strongly aromatic plant, rich in essential oils [29].

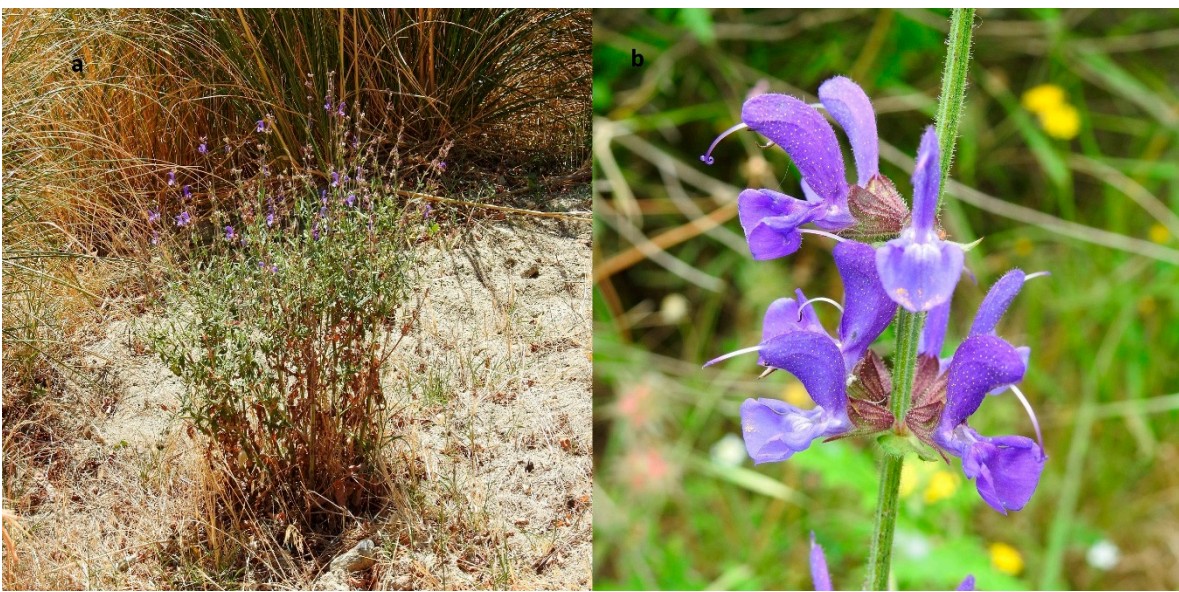

**Figure 1.** *Salvia ceratophylloides* Ard: (**a**) Plant habitat; (**b**) inflorescence in detail, with the bud and open flowers displayed in verticillasters.

## 2.2. Sampling Sites

*S. ceratophylloides* grows in Reggio Calabria (latitude 38°4′, longitude 15°41′), on the hills of the western side of the Aspromonte facing the Strait of Messina (southern Italy, Calabria, Reggio Calabria), at an elevation ranging from 250 to 400 m a.s.l., especially on the cooler slopes with north or northwest exposition [28,30] (Figure 2a). The geology of the area is characterized by layers of loose sand alternating with benches of soft calcarenites of Pliocene origin. The soils have a sandy texture with a basic pH and fall into the group of Calcaric Cambisols [31]. The average temperature in the study area (1971–2000) is approximately 18 °C, with an average annual rainfall of 500 mm concentrated in the autumn and a summer dry period of approximately 5 months [32]. *S. ceratophylloides* grows spontaneously in the habitat of the EEC directive 43/93: "5330 thermo Mediterranean and predesert scrub" subtype "32.23 diss-dominated garrigues." This habitat includes Mediterranean grasslands with *Ampelodesmos mauritanicus* (Poir.) Dur. & Schinz. (Figure 2a), vegetation of the sands with *Artemisia campestris* subsp. *variabilis* (Ten.) Greuter (Figure 2b), or more rarely garrigue with *Cistus creticus* L. (Figure 2c) [17]. The most frequent species growing with *S. ceratophylloides*, in addition to the aforementioned species, are some grasses (*Lagurus ovatus* L., *Avena barbata* Link, *Briza maxima* L., *Hyparrhenia hirta* (L.) Stapf., and *Dasypyrum villosum* (L.) P. Candargy), several dwarf shrubs (*Cistus creticus* L., *Cistus salviifolius* L., *Micromeria graeca* (L.) Benth. ex Rchb., *Thymbra capitata* (L.) Cav., and *Phlomis fruticosa* L.), and some shrubs (*Cytisus infestus* (C. Presl) Guss. subsp. *Infestus* and *Spartium junceum* L.) (Figure 2b). Mostly, they are widespread species in steppic grasslands and in Mediterranean garrigues.

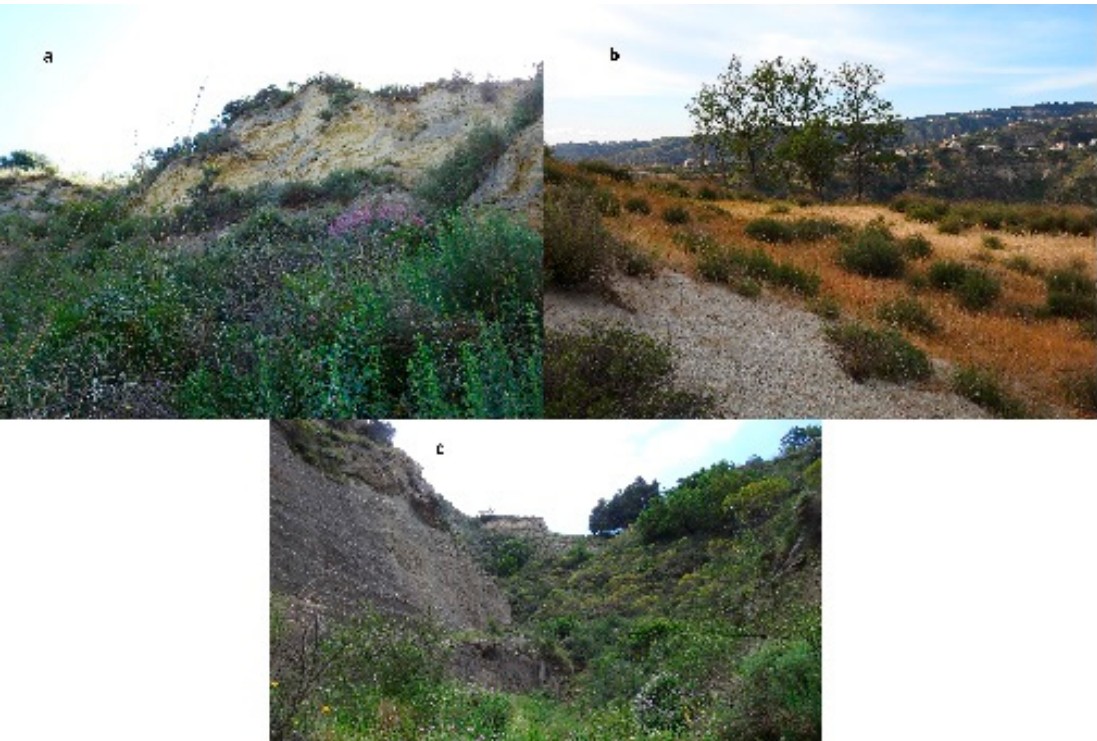

**Figure 2.** Habitat of *S. ceratophylloides*: (**a**) Mediterranean grasslands with *Ampelodesmos mauritanicus* (Poir.) Dur. & Schinz.; (**b**) vegetation of the sands with *Artemisia campestris* subsp. *variabilis* (Ten.) Greuter; (**c**) garrigue with *Cistus creticus* L.

*2.3. Data Collection*

2.3.1. Study Methods and Field Observations

In 2018, in July, mature sage shoots from fully developed plants were collected ($N = 20$) from different sites in the study area. Each plant was observed and many of them (plants not producing inflorescence) displayed emerging holes along the principal axis. In the same year, approximately 200 seeds were collected and stored at laboratory room temperature. The seeds were observed weekly for emerging seed feeders. After these initial reports, we verified the threats posed by herbivores. A more detailed study was carried out in 2019, during which 18 sage sites were discovered and selected as study fields (Table 1). The study sites were within an area of approximately 7 km$^2$ and the minimum distance between the studied sage sites was 375 m. Starting from April until August, the sites were monitored every 15 days. Based on size, we also selected subsites (Table 1), for each of which the sage population structure was analyzed, and all sage plants were recorded and separated as plants in either the reproductive or the vegetative phase [33]. Moreover, we counted both healthy and infested plants; the latter were easily recognizable from June onward as they stopped the development and produced no inflorescence, also showing a yellow/green foliage (Figure 3a). In more advanced infections, it was possible to detect stems with clearly visible holes, particularly present in the basal and middle part of the plant (Figure 3b,c).

**Table 1.** Sampled sites with the presence of *S. ceratophylloides*. For all of the sites, the number of plants with different characteristics were counted.

| Station ID | Subsite | Date | Surface Subsite (m²) | Non-Breeding Plants | Breeding Plants | Total Number of Plants | Infested Plants |
|---|---|---|---|---|---|---|---|
| 1 | 1° | 17.06.2019 | 16 | 29 | 14 | 43 | 5 |
|  | 1B |  | 6 | 5 | 11 | 16 | 3 |
| 2 | 2A | 17.06.2019 | 12 | 2 | 18 | 20 | 18 |
|  | 2B |  | 2 | 0 | 2 | 2 | 2 |
|  | 2C |  | 1 | 0 | 1 | 1 | 1 |
| 3 | 3A | 25.06.2019 | 36 | 15 | 46 | 61 | 35 |
|  | 3B |  | 12 | 0 | 7 | 7 | 4 |
| 4 | 4 | 22.06.2019 | 15 | 1 | 17 | 18 | 0 |
| 5 | 5A | 22.06.2019 | 10 | 5 | 12 | 17 | 0 |
|  | 5B |  | 1200 | 36 | 236 | 272 | 0 |
| 6 | 6A | 19.06.2019 | 80 | 0 | 46 | 46 | 15 |
|  | 6B |  | 265 | 52 | 41 | 93 | 8 |
|  | 6C |  | 168 | 14 | 78 | 92 | 30 |
|  | 6D |  | 1 | 0 | 1 | 1 | 0 |
|  | 6E |  | 6 | 0 | 6 | 6 | 2 |
|  | 6F |  | 1 | 0 | 1 | 1 | 0 |
| 7 | 7A | 22.06.2019 | 3 | 0 | 8 | 8 | 4 |
|  | 7B |  | 25 | 50 | 21 | 71 | 16 |
| 8 | 8 | 22.06.2019 | 8 | 4 | 9 | 13 | 1 |
| 9 | 9 | 22.06.2019 | 10 | 4 | 5 | 9 | 2 |
| 10 | 10 | 22.06.2019 | 120 | 6 | 120 | 126 | 20 |
| 11 | 11A | 19.06.2019 | 1 | 0 | 3 | 3 | 2 |
|  | 11B |  | 175 | 12 | 19 | 31 | 3 |
|  | 11C |  | 8 | 11 | 12 | 23 | 0 |
|  | 11D |  | 9 | 4 | 0 | 4 | 0 |
|  | 11E |  | 9 | 39 | 10 | 49 | 1 |
|  | 11F |  | 35 | 3 | 26 | 29 | 0 |
|  | 11G |  | 2 | 1 | 2 | 3 | 0 |
|  | 11H |  | 1 | 0 | 1 | 1 | 0 |
|  | 11I |  | 15 | 15 | 7 | 22 | 0 |
| 12 | 12A | 17.06.2019 | 50 | 0 | 25 | 25 | 6 |
|  | 12B |  | 3 | 4 | 4 | 8 | 2 |
| 13 | 13 | 17.06.2019 | Extinct | 0 | 0 | 0 | 0 |
| 14 | 14A | 23.06.2019 | 20 | 4 | 15 | 19 | 1 |
|  | 14B |  | 40 | 1 | 8 | 9 | 2 |
| 15 | 15A | 22.06.2019 | 10 | 0 | 55 | 55 | 2 |
|  | 15B |  | 1 | 0 | 5 | 5 | 0 |
| 16 | 16 | 22.06.2019 | 3 | 2 | 16 | 18 | 0 |
| 17 | 17 | 22.06.2019 | 25 | 2 | 10 | 12 | 2 |
| 18 | 18 | 22.06.2019 | 4 | 0 | 3 | 3 | 0 |

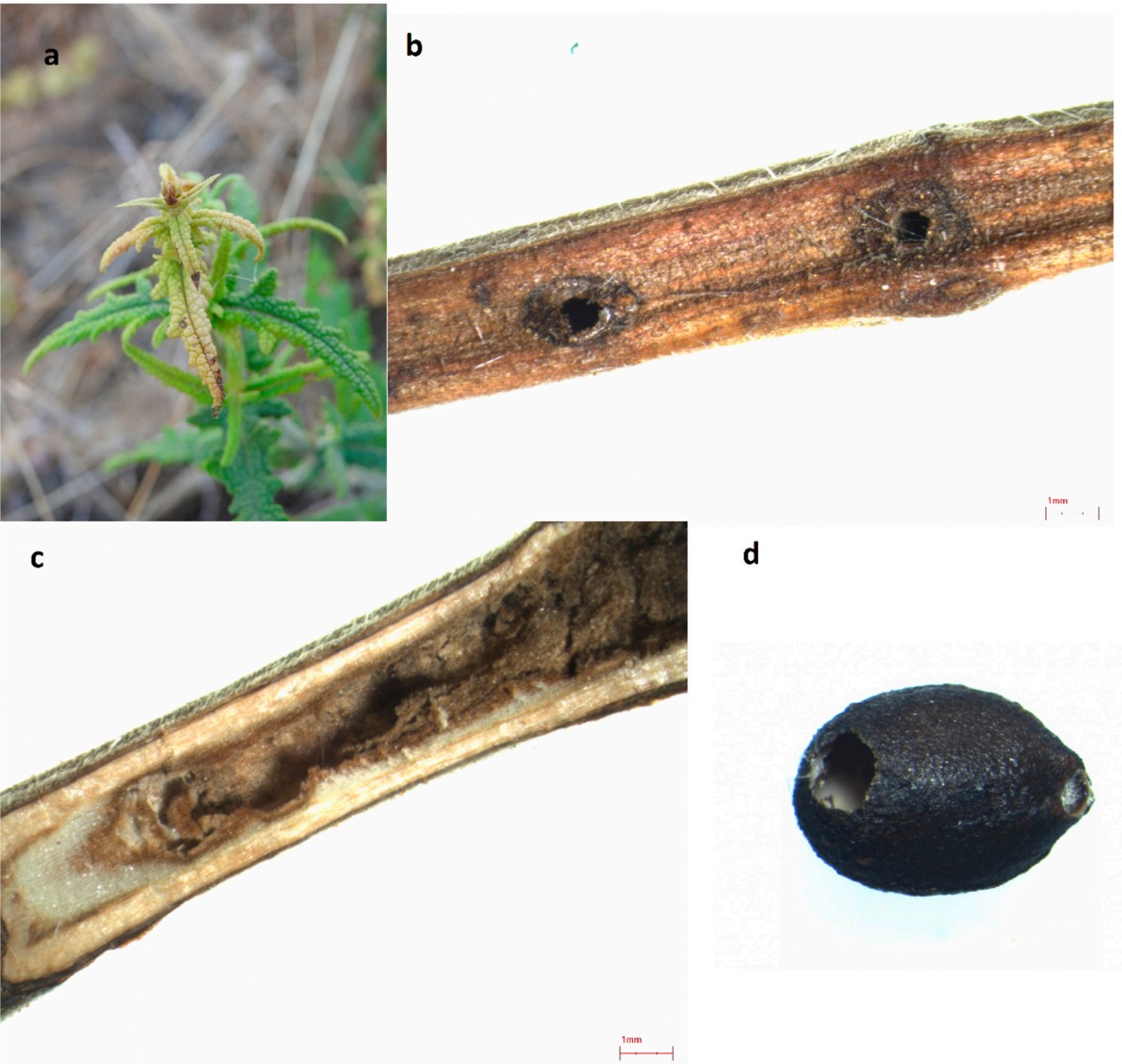

**Figure 3.** (**a**) *S. ceratophylloides* plants with a yellowed apex; (**b**) visible holes on the stem; (**c**) tunnels dug by *Squamapion elongatum;* (**d**) predated sage seed with an exit hole.

### 2.3.2. Borers and Their Natural Enemies

Considering the small number of plants present in all areas, the sampling involved the five sites where the number of sage plants was >40. In these sites, 10 individuals in the incipient flowering phase with some evidence of infestation (yellow/green color) were collected, cleaned of any external presences of insects, and transferred to the laboratory individually inside insect cages. To prevent or reduce evapotranspiration and plant drought stress, the roots were maintained directly in distilled water, both replaced and reviewed, on average, every three days. All of the emerging herbivorous specimens and their parasitoids were removed and preserved in absolute alcohol. The plants in the cage at the end of laboratory trial (November) were examined with a $20\times$ magnification hand lens (Eschenbach® Germany) to search for emerging holes made by insects. For each plant's shoots, the orthogonal section of the branch and stems was measured and used to carry out the study of the herbivorous species.

### 2.3.3. Seed Feeders and Their Natural Enemies

In the last 10 days of June, inflorescences with mature seeds (with at least 80% of seeds formed) were collected in five sites (Table 1). These seeds were separated per site

and observed under a stereoscope; any damaged or shriveled seeds were excluded from further observation. The healthy seed were placed in seed sachets, which contained approximately 200 seeds each (production of one or two adjacent plants) or less, depending on the production per site in some cases ($N$ = 5932 seeds collected). All of the seeds were stored at laboratory temperature (approximately 20 °C) and checked for insect emergence (Figure 3d). Observations were carried out two times a week from the collection date to December. The insect specimens were recorded and separated by site and date. All emerged insects were preserved in absolute alcohol for storage.

### 2.3.4. Herbivores and Natural Enemies' Identification

Each specimen that emerged from the plants and seeds was observed under a stereomicroscope (Olympus SZX9, Tokyo, Japan) at magnifications from 20× to 60×. Taxonomic identification was made using the dichotomous key for the Apionidae beetles [34–37], while for the seed calcidids and parasitoids, the dichotomous keys of Baur et al. [38–44] were used.

### 2.4. Data Analyses

For the statistical analyses, we used the total number of sage plants detected. Differences among the populations of plants in the reproductive and vegetative phases were analyzed using Generalized Linear Model with the dependent variable (number of plants infested) adjusted to a binomial negative distribution with a log link function to model responses. The relationship among the plants infested, the total number of plants, and the reproductive phase at the date of monitoring were analyzed using Spearman's correlation coefficients. The proportion of infested plants at different sites was compared using a Kolmogorov–Smirnov (K-S) test of the equality of distributions. The relationships between the numbers of emerging holes and the mean diameter of each plant branch were interpreted using nonparametric analysis based on Spearman's correlation coefficients.

The parasitization of the seeds (number of seeds in which an herbivore or its parasitoid emerges) was evaluated at different sites. Fisher's exact test in a contingency table (two-tailed, $p < 0.05$) was used to assess the significance of any differences of the species that emerged at different sites. We used SPSS v. 23 [45] for all data analyses and Sigmaplot 13.0 [46] to produce graphs. All data are expressed as untransformed mean values ± standard errors (SEs).

### 2.5. Molecular Identification

DNA Extraction, Amplification, and Sequencing

The species involved in this study were collected and stored individually in Eppendorf tubes with absolute ethanol at −20 °C. Rather than grinding the specimens, total genomic DNA was extracted from 10 samples of each species, following the protocol described in Vono et al. [47]. DNA extraction was performed on individual insects in 5 μL of proteinase K (20 mg/mL) and 80 μL of 5% Chelex 100 suspension, and then incubated at 55 °C for 1 h. Proteinase K was then inactivated at 100 °C for 8 min. The supernatant containing the DNA was removed after centrifugation and stored at −20 °C. Three genes were sequenced: the mitochondrial cytochrome c oxidase subunit I (COI), and ribosomal gene, and the expansion segment D2 of the 28S ribosomal subunit (28S-D2). The polymerase chain reaction (PCR) cycle (thermocycle conditions) and primers used to amplify a fragment of mitochondrial COI and a fragment of 28S-D2 ribosomal are reported in Table 2.

**Table 2.** Primer sequences with the relative amplification program of the mitochondrial cytochrome c oxidase subunit I (COI) and the expansion segment D2 of the 28S ribosomal subunit (28S-D2) genetic regions for molecular characterization of insect species.

| Name | Sequence 5′–3′ | Fragment | Source | PCR Cycle | | |
| --- | --- | --- | --- | --- | --- | --- |
| | | | | T (°C) | Time | N of Cycle |
| LCO-1490 | GGTCAACAAATCATAAAGATATTGG | COI | [48] | 95 | 1′ | |
| | | | | 94 | 30″ | |
| | | | | 48 | 1′30″ | 40 |
| HCO-2198 | GTAAATATATGRTGDGCTC | | | 72 | 1′ | |
| | | | | 72 | 7″ | |
| D2F | CGTGTTGCTTGATAGTGCAGC | 28S-D2 | [49] | 95 | 3′ | |
| | | | | 94 | 45″ | |
| | | | | 55 | 45″ | 35 |
| D2R | TTGGTCCGTGTTTCAAGACGGG | | | 72 | 1′ | |
| | | | | 72 | 7″ | |

PCR = polymerase chain reaction.

For gene amplifications, PCR was performed on a BioRad thermocycler using 20 μL of the reaction volumes, consisting of 1× Promega PCR buffer (containing MgCl$_2$), 0.2 mM each of dNTP, 0.25 μM of each primer, 10 mg/mL of bovine serum albumin, 1.5 units of Go*Taq* G2 DNA polymerase (Promega Italia, Milan, Italy), and 2 μL of the DNA template. The PCR products were checked on a 1.2% agarose gel stained with GelRED® (Biotium, Fremont, CA, USA), and then visualized and photographed under ultra-violet (UV) light. Sequences were trimmed and aligned manually in MEGA v. 7 and were virtually translated into the corresponding amino acid chain to detect frame-shift mutations and stop codons, using EMBOSS Transeq (https://www.ebi.ac.uk/Tools/st/emboss_transeq/) (accessed 21 April 2020) [50]. Edited sequences were checked against the GenBank database, and then submitted to the GenBank database under the accession numbers reported in Supplementary Table S1.

## 3. Results

### 3.1. Field Observations

The average surface area of the sage of each site was 60.2 ± 30.54 m$^2$, with a maximum surface area in one site reaching 1200 m$^2$. Overall, the total area covered by the sage was 2408 m$^2$. The proportion of breeding plants in all sites at the time of sampling was 0.7716.

The infested plants were positively related to the reproductive phase (i.e., reproductive plants (RPs)) ($r_s$ = 0.517, $n$ = 39, $p < 0.001$) and not to the vegetative phase (i.e., growing plants (GPs)) or sites (Table 3).

There was no relationship between the infested and total plants in each site ($r_s$ = 0.224, $n$ = 40, $p$ = 0.164). In all sites, the average proportion of infested plants not producing inflorescence was 0.2142. The distribution of the proportions of the infested plants was not uniform in the different sites (K-S tests = 3.082; $n$ = 39; $p < 0.001$) with vast differences.

The number of plants in the laboratory that showed emergence holes was 8/10 collected. The number of holes varied from a minimum of two up to a maximum of 23 holes per plant. The average diameter of the shoot at the exit point was 4.55 mm ($n$ = 68), while the average diameter of the shoots without holes was 2.31 mm ($n$ = 14). For the plants examined, the association between the number of holes and the average diameter of the shoot showed a positive relationship ($r_s$ = 0.81, $n$ = 13, $p < 0.001$). A principle axis or secondary stem with a diameter less than 2.30 mm was not infested.

**Table 3.** Generalized linear model with negative binomial distribution results evaluating the effect of different variables on the plants infested [a].

| Response Variable | Source | Df | Wald Chi-Square | p |
|---|---|---|---|---|
| | Intercept | 1 | 0.91 | 0.340 |
| | Site | 12 | 19.34 | 0.081 |
| Plants infested | Reproductive plants (RPs) | 1 | 12.04 | 0.001 |
| | Growing plants (GPs) | 1 | 0.50 | 0.468 |
| | RP * GP | 1 | 0.392 | 0.081 |

[a] The AIC (Akaike's information criterion) value was −500.74. The likelihood ratio (chi-square) was 71.02 (df = 20; *p* < 0.001).

### 3.2. Complex of Herbivores and Their Natural Enemies on Salvia ceratophylloides Plants

The complex of herbivores and their natural enemies is depicted in Figure 4.

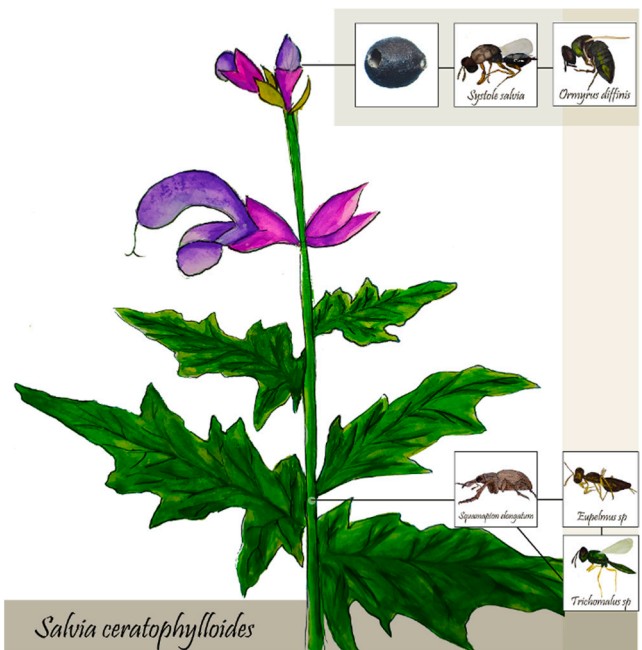

**Figure 4.** Complex of herbivores and their parasitoids present on *S. ceratophylloides*. Plants attacked by *S. elongatum* do not produce inflorescences and become visibly chlorotic. Top: Seed with hole and seed feeder species *Systole salviae* and its parasitoid *Ormyrus diffinis*. Bottom: *S. elongatum* and its parasitoids *Trichomalus* sp., *Eupelmus* sp.

The relationships among the species were well defined, with different species that emerged from the sage plant. They were *S. elongatum* (Germar, 1817) (Coleoptera Apionidae) (Supplementary Figure S1) (emerged *n* = 15) and its parasitoids *Trichomalus* spp. Thomson sp. 1878 (Hymenoptera, Chalcidoidea, Pteromalidae) (emerged *n* = 20), *Eupelmus vesicularis* (Retzius, 1783), and *E. muellneri* Ruschka, 1921 (Hymenoptera, Chalcidoidea, Eupelmidae) (emerged *n* = 4).

### 3.3. Salvia ceratophylloides Seed Herbivores and Their Natural Enemies

The number of seeds harvested was different in relation to the sites (3B = 70; 6A = 3726; 11E = 177; 14A = 1937). In total, 113 adult insect specimens (3.41% of seeds) emerged during summer–autumn (Figure 5).

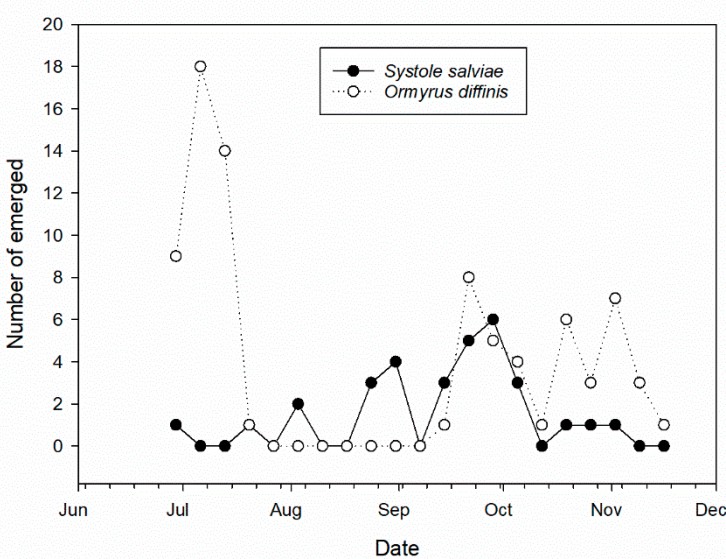

**Figure 5.** Emergence of *S. salvia* and *O. diffinis* during the summer and autumn seasons.

The percentage of seeds attacked by the seed feeders was different in relation to the site, with the minimum in site 6 (0.86%) and the maximum in site 3 (5.71%). Two insect species emerged from the sage seeds: *Systole salviae* (Zerova, 1968) (Hymenoptera, Chalcidoidea, Eurytomidae) and *Ormyrus diffinis* (Fonscolombe, 1832) (Hymenoptera, Chalcidoidea, Ormyridae). The species that emerged were different among the sites, with a preponderance of *O. diffinis* in site 14 and the absence of *S. salviae* emergence in site 11 (Figure 6).

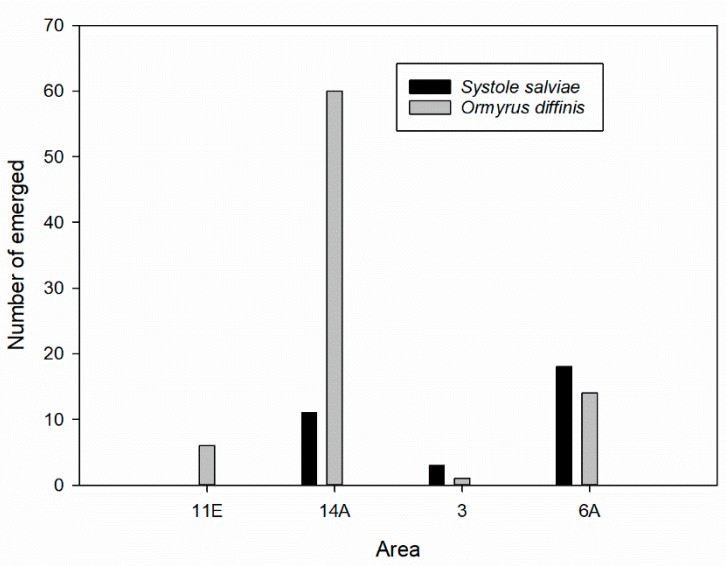

**Figure 6.** Species emerged from *S. ceratophylloides* seeds from the four different sites. There were significant differences in the proportion of the two emerged species (Fisher's exact test = 20.75; $N = 113$; $p < 0.001$).

*3.4. Molecular Identification of the Species*

Molecular analysis was performed in order to confirm the identity of *S. elongatum*, *S. salviae*, *O. diffinis*, and *Tricomalus* spp. Our DNA extraction method proved effective for all species belonging to different orders, and a useful amount of genetic material was extracted to be used in the PCR. For each sample, two genomic regions were amplified. The polymerase chain reaction of *mt*-COI produced fragments between 422 and 622 bp.

The nucleotide composition of the *S. elongatum* sequences was T(U) = 29.6%, C = 15.6%, A = 38.4%, and G = 16.4%. For *S. salviae,* the nucleotide composition was T(U) = 29.7%, C = 14%, A = 44.8%, and G = 11.5%; for *O. diffinis*, it was T(U) = 28.9%, C = 13.6%, A = 45.4%, and G = 12.1%; and for *Trichomalus* spp., it was T(U) = 32.7%, C = 12.8%, A = 41.7%, and G = 12.8%. In all species, the average A + T content was high (mean value = 72.8%), which is in agreement with values for insects in general [48,51]. There were no molecular intraspecific differences in the samples analyzed. For fragment 28S-D2, a highly conserved region, with sequences of 371–555 bp in length, was obtained.

## 4. Discussion

Despite its endemicity and conservation importance in the Italian territory, not much is known about the performance of *S. ceratophylloides* in the area of occupancy. *S. ceratophylloides* populations are threatened by the anthropic activities related to their occurrence near cities, and these novel results highlight a new serious threat to the species primarily linked to *S. elongatum* distributed across all sites. The genus *Squamapion* Bokor, 1923 is distributed in the Palearctic and Afrotropical regions and includes 33 species [52]. These are mono- or oligophagous herbivores feeding on species of the family Lamiaceae, mainly of the genera *Salvia*, *Thymus*, *Thymbra*, *Mentha*, *Origanum*, *Prunella*, and *Saccocalyx*. Their larvae burrow tunnels inside roots or stems, occasionally causing galls [37]. *S. elongatum* is a species inhabiting the southern and central part of Europe, as well as East Asia and Algeria, and it colonizes lowland and sub-montane areas [53]. This species is characteristic of xerothermic grasslands, where it feeds on plants of the genus *Salvia—S. pratensis* and *S. nemorosa* [54]. Other studies prior to this have displayed an overall picture of herbivorous species that affect sage inflorescence (e.g., *S. sclareoides* Brot. [9,55]), but no such striking effect of herbivores has been highlighted in *Salvia* species. Despite the notable spread of the herbivorous beetle, the species is not reported for the most widespread and cultivated sage (*Salvia officinalis*) [56]. Hendrich et al. [57] reported the molecular characterization of the COI gene region performed on samples of *S. elongatum* collected in Germany. The nucleotide sequence of the *mt*-COI gene obtained in this study in order to confirm the identity of the species was compared with the sequences found in GenBank (https://blast.ncbi.nlm.nih.gov/Blast.cgi) (accessed 25 April 2020) [58]). The comparison showed that the percentage of identity between the samples used in this study and those in Germany amounted to 92.75%. This value must be considered valid because the records relating to *S. elongatum* in the database were limited to a few accessions. To date, most published DNA barcoding studies dealing with insects have focused on Hymenoptera and Lepidoptera, while few recent studies have examined the hyperdiverse Coleoptera [59,60]. Apparently, the genetic differentiation among populations has a complex background and may involve factors such as local adaptation and founder effects [61].

Hairston et al. [62] argued that observations of herbivores severely depleting green plants are rare. Wetzel et al. [63] highlighted that plants can contribute to the suppression of herbivorous populations via variable nutrient levels, whereas Koussoroplis et al. [64] considered the fundamental role of plant trait variability on herbivorous populations in natural important settings; however, the real threat of insect herbivores for different plant species is unsolved. Solving the numerous challenges pertaining to this threat can help us understand how herbivores interact with host plants and how changes in abiotic factors influence and constrain the effects of herbivores on their host [6]. For several circumstances and in many cases, the threat for plants species is associated with the introduction of invasive herbivorous species capable of conditioning, for example, reproduction or vitality [65,66]. Although *S. ceratophylloides* is a strong producer of essential oils, *S. elongatum* seems to affect the vitality and reproduction of species. We do not know the effects of this herbivore on, or its relationships with, the second autumn flowering. A secondary flowering is recorded for many species as a display of initially continuous flowering or because of damage [67]. Determining the cause of this second flowering in *S. ceratophylloides* is necessary in order to exclude a flowering related to the herbivores'

damage. The number of emerging holes for *S. elongatum* in each plant was related to the average diameter of plant stems with increased digging of tunnels according to the diameter size. According to the available literature, the period of occurrence of adults of *S. elongatum* is April [54], but in our study areas, the species emerged in June and on the first day of July. Chalcid wasps such as *Trichomalus* and secondary *E. vesicularis* and *E. muellneri* parasitized larvae of this apionid. To the genus *Trichomalus* belong widely diffused species, some of which have good potential for the biological control of other species of beetles such as the cabbage seedpod weevil (CSW), *Ceutorhynchus obstrictus* Marsham (Coleoptera: Curculionidae) [67]. *E. vesicularis* and *E. muellneri* have been detected as parasitoids of *Ruguloscolytus mediterraneus* (Eggers, 1922) (Coleoptera: Scolytidae) and *Sphenoptera davatchii* Descarpentries, 1960 (Coleoptera: Buprestidae) [68]. European species of *Eupelmus* (Macroneura Walker) are cosmopolitan and extremely polyphagous species [69–71].

In general, the amount of insect-feeding seed damage is low [72]. Occurrences of Eurytomid wasps (Hymenoptera: Eurytomidae) in seeds are amply documented for different species [73], and on sage seeds, we confirmed and highlighted the seed feeder *S. salviae* for this sage species. Previously, other finds on *S. pratensis*, *S. glutinosa* L., *S. scabiosifolia* Lam., and probably *S. grandiflora* Etlin. [41–44] have highlighted the emergence of *S. salviae*, while no *O. diffinis* has been reported; our case is the first report regarding a *Systole–Ormyrus* interaction. Based on available literature, *O. diffinis* plays the role of parasitoid for other species of *Ormyrus* sp. [74–77]. This species is also reared from the galls of *Neaylax alvia* and *N. nemorosae* (Hymenoptera Cynipidae) on different species of sage and from those of *Rhodus cyprius* (Hymenoptera Cynipidae) on *Salvia triloba* (Lamiaceae) [74]. *O. diffinis* is widespread and has only recently been reported in Iran [68]. In any case, for *O. diffinis*, the possible role in regulating the *S. salviae* population, as well as other better-known pest species [78], remains unknown.

In this study, considering the obtained data, we did not explore if this sage species exhibits a masting behavior and if environmental factors drive masting and the emergence of herbivores [79]. Espelta et al. [80], in two mid-term studies, showed, for example, that acorn production and predation are highly variable across years, while the abundance of adult weevils is positively related to autumn rainfall and to the number of infested acorns the previous years. Further investigations in our area are in progress in order to understand if the production of seeds in *S. ceratophylloides* follows the reproductive economy through seed predation for selecting the synchronous production of massive and null seed crops in this species.

## 5. Conclusions

Investigations will be carried out in order to evaluate and quantify in more detail the quantitative relationships between the abundance of *S. elongatum* larvae and inflorescence and seed production in *S. ceratophylloides*. We suppose that in reproductive plants, a low presence of herbivorous larvae could also influence the production of pollen or the abundance of seeds. Notwithstanding that the attack of apionids affects the internal tissues of the stem, damage to inflorescence formation was detected. The data acquired indicate that the density of the herbivore in the sage diffusion area does not represent a factor for quantitative regulation of flowering, but it is rather able to condition species survival. This interaction between host and established herbivores promotes a behavioral model of the pest populations affecting the reproduction of this sage species, which helps to explain their ecological relationship with the host plant, defining the real harmfulness, and to better undertake any phytoiatric decision and any plant species conservation activity. Conversely, the role of seed feeders appears to be very different, which seems more limited, and the relationship with sage is to be considered tolerable under natural conditions. Generally, seed predators should have little influence on many host plant populations, and further investigations are necessary to verify whether conjunct attack of both herbivores the same year causes a sharp reduction in sage in the area indicated.

**Supplementary Materials:** The following are available online at https://www.mdpi.com/1424-281 8/13/1/33/s1, Table S1: Information about *Squamapion elongatum* and accession numbers related to the gene sequences of samples analysed. Figure S1: Dorsal and lateral view of *Squamapion elongatum*.

**Author Contributions:** C.P.B. conceived and designed the experiments; C.P.B. and G.V. performed the experiments, analyzed the data, wrote the paper, and contributed reagents/materials/analysis tools; V.L.A.L. collected the plants and materials, took the photos, and wrote and reviewed the paper; R.M., G.S. and C.M.M. collected the plants and materials and reviewed drafts of the paper. All authors have read and agreed to the published version of the manuscript.

**Funding:** The research was partly supported by the Internal Grant Agency (FFABBR, funding basic research activities 2017) assigned to C.P. Bonsignore.

**Institutional Review Board Statement:** Not applicable.

**Informed Consent Statement:** Not applicable.

**Data Availability Statement:** The data presented in this study are reported in the tables and supplementary materials. Data not shown in tables are available on request from the corresponding author. The data are not publicly available due to ongoing longitudinal analysis.

**Acknowledgments:** The authors wish to thank Michele Calabrò, who accompanied them on field collection of *S. ceratophhylloides*. Hannes Baur, Krzysztof Pawlega, Hosseinali Lotfalizadeh, Lucian Fusu, José-Luis Aldrey, and Irinel E. Popescu are thanked for information regarding the identification of the detected species. The authors thank Chiara Valenti for drawing Figure 4 and Giusi Vizzari for providing the lab facilities for the experiments to be carried out in. All photos in the figures were taken by Valentina Lucia Astrid Laface. The authors also thank the three anonymous reviewers and the academic editor whose comments helped improve this manuscript.

**Conflicts of Interest:** The authors declare no conflict of interest. The funders had no role in the design of the study; in the collection, analyses, or interpretation of data; in the writing of the manuscript, or in the decision to publish the results.

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
