# Peer review of "Threats Posed to the Rediscovered and Rare Salvia ceratophylloides Ard. (Lamiaceae) by Borer and Seed Feeder Insect Species"

_diversity, doi:10.3390/d13010033_

Round 1

Reviewer 1 Report

I did not fully understand the methods of this paper. It would be helpful to clearly explain what was measured, how was it measured and why was it measured? Also, the paper will benefit from some language editing. Revise word choice and sentences completeness. See some comments below (they are not extensive):   

Lines 18-19: "plants in site" do you mean "in situ" observations?

Line 20: "whatever" do you mean "which"?

Line 23: "Threats sage" do you mean threat? 

Lines 53-55: This sentence is not clear

Lines 118-119: This sentence is not clear

Lines121-122: "The seeds were weekly observed for emerging" for emerging what? 

Line 124: I did not see table 1. 

Line 127: "For each site, we detected soil surface and analyzed the population stage structure" I'm not sure what the authors are trying to say here. 

Lines 127-128: "All sage plants were recorded and separated in plants in reproductive phase and plant in growing [33]" Do you mean reproductive and vegetative stages? 

Lines 146-148: "On each shoots the orthogonal section of the branch and stems interested or not in the hole emergence have been measured in which it was possible to carry out the study of the herbivores species." I'm not sure what the authors are trying to say here.

Line 150: "decade" decade means 10 years, this is very confusing. 

Line 155: "All seeds were stored at laboratory temperature (about 20 °C) and checked for emerged (Figure 3d)" For emerged what? 

Lines 155-157: "The observations on the emerged from seeds were carried out two times a week from the collection date to December." For emerged what? 

There are many more sentences throughout the manuscript that I found difficult to understand. It would be helpful to revise word choice and sentences completeness. 

Author Response

Reviewer 1

Comments and Suggestions for Authors

I did not fully understand the methods of this paper. It would be helpful to clearly explain what was measured, how was it measured and why was it measured? Also, the paper will benefit from some language editing. Revise word choice and sentences completeness. See some comments below (they are not extensive):   

[A]We thank you for your valuable suggestions that have improved the quality of the manuscript. We have accepted  all of your advices. The entire MS has been revisited according to the suggestions and in particular we improve the readability of ms. The MS has been corrected in its different parts in order to make it more comprehensible for a linear reading and is now less verbose. The objectives that we set ourselves to reach the goal of the research were reformulated in order to make the main objective more comprehensible

We have checked of the paper for English language according to MDPI editing service.

Lines 18-19: "plants in site" do you mean "in situ" observations?

[A] We changed words”.

Line 20: "whatever" do you mean "which"?

[A] Yes, we change word.

Line 23: "Threats sage" do you mean threat? 

[A] Yes, we changed word.

Lines 53-55: This sentence is not clear

[A] The sentence was modified to better understanding

Lines 118-119: This sentence is not clear

[A] The sentence was modified to better understanding

Lines121-122: "The seeds were weekly observed for emerging" for emerging what? 

[A] "…for emerging insect seed feeders” The sentence was modified to better understanding

Line 124: I did not see table 1. 

[A] We apologies about this. Now table 1 is correctly added.

Line 127: "For each site, we detected soil surface and analyzed the population stage structure" I'm not sure what the authors are trying to say here. 

[A] We reformulated sentence. It has partly been simplified, as it was verbose and unclear.

Lines 127-128: "All sage plants were recorded and separated in plants in reproductive phase and plant in growing [33]" Do you mean reproductive and vegetative stages? 

[A] Yes we mean reproductive and vegetative stages and we change sentence accordingly.

Lines 146-148: "On each shoots the orthogonal section of the branch and stems interested or not in the hole emergence have been measured in which it was possible to carry out the study of the herbivores species." I'm not sure what the authors are trying to say here.

[A] The sentence was simplified.

Line 150: "decade" decade means 10 years, this is very confusing. 

[A] We removed word and we change with “the last 10 days” to better understand.

Line 155: "All seeds were stored at laboratory temperature (about 20 °C) and checked for emerged (Figure 3d)" For emerged what? 

[A] We added words in the sentence.

Lines 155-157: "The observations on the emerged from seeds were carried out two times a week from the collection date to December." For emerged what? 

[A] We changed the sentence adding more information.

There are many more sentences throughout the manuscript that I found difficult to understand. It would be helpful to revise word choice and sentences completeness. 

[A] We have modified several sentences in the text and also revised the words entered in the MS. Finally the MS underwent review for the English language.

Reviewer 2 Report

Dear authors.

This manuscript is very interesting and acceptable for publication after minor revision.

Just some comments:

Intoduction.

line 44. round bracket is missing (reduction....)?

line 85. in full name after the point and in italics. (no S. ceratophylloides but Salvia ceratophylloides)

line 88. standardize bibliography, No (Spampinato et al. 2018 but [29]?

line 94. standardize in line 56 "Aspromonte Massif" has been used

line 102. "32.23 dissdominated garrigues" it is generally written as "...Diss-dominates..."

line 104. remove space in fig 2 b

lines 124-126. (Table 1) is missing in the pdf to be able to check it

line 155. remove bold in "(Figure 3d)"

line 235. replace affinis with diffinis

line 247. In the figure 5, replace affinis with diffinis

line 273. in full name after the point "Salvia ceratophylloides"

line 279. in full name after the point "Squamapion elongatum"

Author Response

Reviewer 2

Comments and Suggestions for Authors

Dear authors.

This manuscript is very interesting and acceptable for publication after minor revision.

First, we thank you  the reviewers for their remarks on our manuscript. We really appreciate that you consider that our manuscript is technically sound, well structured and that the data support the conclusions.

Just some comments:

Introduction.

line 44. round bracket is missing (reduction....)?

[A] We added the bracket.

line 85. in full name after the point and in italics. (no S. ceratophylloides but Salvia ceratophylloides)

[A] We added bracket full name after the point.

line 88. standardize bibliography, No (Spampinato et al. 2018 but [29]?

[A] We standardized reference in this sentence.

line 94. standardize in line 56 "Aspromonte Massif" has been used

[A] We standardized reference in this sentence.

line 102. "32.23 dissdominated garrigues" it is generally written as "...Diss-dominates..."

[A] We corrected dissdominated garrigues in Diss-dominated...  standardized reference in this sentence.

line 104. remove space in fig 2 b

[A] We corrected.

lines 124-126. (Table 1) is missing in the pdf to be able to check it

[A] We apologies about this. Now table 1 is correctly added.

line 155. remove bold in "(Figure 3d)"

[A] We corrected.

line 235. replace affinis with diffinis

[A] We replaced with diffinis.

line 247. In the figure 5, replace affinis with diffinis

[A] We replaced with diffinis. Also we corrected word in Figure 5.

line 273. in full name after the point "Salvia ceratophylloides"

[A] We added the full name after the point.

line 279. in full name after the point "Squamapion elongatum"

[A] We added the full name after the point.

Reviewer 3 Report

The manuscript entitled "Threats Posed to Rediscovered and Rare Salvia ceratophylloides Ard. (Lamiaceae) by Borer and Seed Feeders Insect Species" provides new useful insights about plant-insect interactions related to a strict endemic sage species.
The title is properly conceived and well reflects the manuscript contents.
The abstract clearly summarizes the manuscript contents, the same is for the proposed keys words.
The manuscript is overall well-structured: the introduction provides a clear background of the general topic and the status of Salvia ceratophylloides in particular. Methodological approaches are diversified and properly structured. Results and discussion are balanced and well supported. The reference list is comprehensive.

There were however lots of oversights and mistakes.
English language needs a careful revision, as many grammatical and syntactical errors were detected and highlighted directly in the text (but I cannot ensure I have seen them all).
Insect nomenclature throughout the mascript must be checked and standardized according to the International Code of Zoological Nomenclature (I found and modified some inconsistencies both in species name and authors, but I am not a zoologist).
A relevant inadvertence was missing the table listing the sample provenance (Table 1). This table with related specifications is mandatory for publication.
The editing of reference list must be also carefully revised.
Please find all my comments and corrections in the attached pdf file

Overall, some minor changes are needed.

Author Response

Reviewer 3

The manuscript entitled "Threats Posed to Rediscovered and Rare Salvia ceratophylloides Ard. (Lamiaceae) by Borer and Seed Feeders Insect Species" provides new useful insights about plant-insect interactions related to a strict endemic sage species.

We thank you for your remarks on our manuscript. We really appreciate that the reviewers (2,3) consider that our manuscript is technically sound, well structured and that the data support the conclusions.

The title is properly conceived and well reflects the manuscript contents.
The abstract clearly summarizes the manuscript contents, the same is for the proposed keys words.
The manuscript is overall well-structured: the introduction provides a clear background of the general topic and the status of Salvia ceratophylloides in particular. Methodological approaches are diversified and properly structured. Results and discussion are balanced and well supported. The reference list is comprehensive.

There were however lots of oversights and mistakes.
English language needs a careful revision, as many grammatical and syntactical errors were detected and highlighted directly in the text (but I cannot ensure I have seen them all).
Insect nomenclature throughout the mascript must be checked and standardized according to the International Code of Zoological Nomenclature (I found and modified some inconsistencies both in species name and authors, but I am not a zoologist).
A relevant inadvertence was missing the table listing the sample provenance (Table 1). This table with related specifications is mandatory for publication.
The editing of reference list must be also carefully revised.
Please find all my comments and corrections in the attached pdf file

[A] We really appreciate that you consider our manuscript is technically sound, well structured and that the data support the conclusions. We thank you for the suggestions you have provided and for the corrections indicated along the text of the MS. We also apologize for making mistakes such as failure to enter table 1. The latter has been entered correctly. The insect nomenclature has been revised according to the International Code of Zoological Nomenclature.

We have checked of the paper for English language according to MDPI editing service.

 Overall, some minor changes are needed.

The revisions reported on the pdf are highlighted in revised MS.

Round 2

Reviewer 1 Report

Line 43: the word “basically” is unnecessary

Line 127:  decade means 10 years, please correct this.

Line 164: Replace “Examined by” for “examined with”

Author Response

Letter to the Editor and Reviewers:

11 January 2021

Ref.:  Ms. No. diversity-1023585(3)
Journal: Diversity

Letter to the Academic Editor and Reviewers:

Dear Sirs,

We gratefully thank you for your invitation to resubmit the revised version of our paper (diversity-1023585) entitled « Threats Posed to the Rediscovered and Rare Salvia ceratophylloides Ard. (Lamiaceae) by Borer and Seed Feeder Insect Species. » by CP Bonsignore et al.

First, we thank the reviewer for his remarks on our manuscript. We really appreciate that the reviewer consider our manuscript is technically sound, well structured.

We have considered all reviewer remarks in the following manner. Our authors answers [A] are below in bold characters letter to Editor and reviewers. All changes on MS are reported in file word format and the new words and sentence added in track change.

Comments and Suggestions for Authors

Line 43: the word “basically” is unnecessary

[A] The word has been removed;.

Line 127: decade means 10 years, please correct this.

[A] We have corrected sentence and the word has been removed;

Line 164: Replace “Examined by” for “examined with”

[A] We have changed word in sentence.

Best Regards 

C. P. Bonsignore